# Vocational rehabilitation for people with multiple sclerosis: A systematic scoping review of international evidence

Carlotta Gualco[1]*, Erica Grange[2,3], Federica Rotondi[1], Marco Salivetto[1], Elena Pignattelli[1], Tommaso Manacorda[4], Maria Grazia Grasso[5], Giorgia Presicce[5], Matilde Inglese[6,7], Lorenza Nasone[6], Paolo Durando[8,9], Guglielmo Dini[8,9], Benedetta Persechino[10], Giampaolo Brichetto[2,3], Michela Ponzio[1]

1 Public Health Research Area, Italian Multiple Sclerosis Foundation (FISM), Genoa, Italy, 2 NeuroBRITE Research Center, Italian Multiple Sclerosis Foundation (FISM), Genoa, Italy, 3 Department of Informatics, Bioengineering, Robotics and Systems Engineering (DIBRIS), University of Genova, Genova, Italy, 4 Italian Multiple Sclerosis Association (AISM), Genoa, Italy, 5 Multiple Sclerosis Unit, IRCCS S. Lucia Foundation, Rome, Italy, 6 Department of Neurosciences, Rehabilitation, Ophthalmology, Genetics, Maternal and Child Health (DINOGMI), University of Genoa, Genoa, Italy, 7 IRCCS Azienda Ospedaliera Metropolitana, Genoa, Italy, 8 Department of Health Sciences, University of Genoa, Genoa, Italy, 9 Occupational Medicine Unit, IRCCS Ospedale Policlinico San Martino, Genoa, Italy, 10 Italian Workers' Compensation Authority (INAIL), Rome, Italy

* gualco.carlotta95@mail.com

## Abstract

### Introduction

People with multiple sclerosis (pwMS) may encounter challenges in their professional lives, due to a combination of environmental and individual factors. According to Escorpizo et al., 2011 framework, Vocational rehabilitation (VR) aims to optimise job participation, providing support in the job access, retention and in the return to work for people with disability. However, the corpus of research on VR for pwMS is poor. This scoping review aims to map the available literature on VR interventions for pwMS, summarising their characteristics, study designs, implementation features, feasibility, and stakeholders' perspectives.

### Methods

Following the Joanna Briggs Institute (JBI) and the PRISMA-ScR guidelines, seven databases were searched up to October 2025: PubMed, SCOPUS, PsycInfo, CINAHL, Google Scholar, OT Seeker (University of Queensland), and the Physiotherapy Evidence Database (PEDro). Studies were eligible if they were related to VR interventions for pwMS, focused on job access, return, or retention and if they were primary articles. Data were extracted and synthesised following the Population–Concept–Context (PCC) framework.

**Data availability statement:** All relevant data are within the manuscript and its Supporting information files.

**Funding:** This study was funded by the Italian Workers' Compensation Authority (INAIL), in the framework of the BRIC 2022: "RiaL SM" project (Bando BRIC 2022_ID 31). However, the funders had no role in study design, data collection and analysis, decision to publish, or preparation of the manuscript.

**Competing interests:** The authors have declared that no competing interests exist.

**Abbreviations:** MS, Multiple Sclerosis; pwMS, people with Multiple Sclerosis; VR, Vocational Rehabilitation; RCT, Randomized Controlled Trial; OSF, Open Science Framework; EDSS, Expanded Disability Status Scale; PRISMA-ScR, Preferred Reporting Items for Systematic Reviews and Meta-analyses extensions for Scoping review; JBI, Joanna Briggs Institute; RA, Reasonable accomodations; RR, Relapsing-Remitting; NA, Not applicable; NR, Not reported; OT, Occupational therapist.

## Results

Out of 2,360 records, 28 articles describing 28 distinct VR interventions were included. Studies were published between 1996 and 2025, mostly from Western countries. Designs ranged from descriptive to randomized trials, with an increasing number of interventional and feasibility studies in recent years. The 61% of the interventions were multi-dimensional delivering a combination of rehabilitation, educational, and reasonable accommodation services. PwMS highlighted the importance of empathetic and individualized approaches, symptom management, and legal counselling as key elements in VR interventions, while logistical, personal and health issues were barriers to participation. Overall, interventions were considered feasible and acceptable.

## Conclusions

This is the first comprehensive overview of VR interventions for pwMS, outlining a progressive shift toward multidisciplinary and goal-oriented approaches over time. Despite promising feasibility and stakeholder satisfaction, further rigorous trials are needed to evaluate effectiveness and inform evidence-based implementation of VR programmes in diverse contexts.

## Introduction

Multiple sclerosis (MS) is a chronic disease of the central nervous system characterised by autoimmune and neurodegenerative processes [1]. MS presents a wide range of physical, cognitive and psychological symptoms that may have a detrimental effect on an individual's work capacity [2–4]. This may lead to challenges in accessing employment, returning to work after diagnosis, and job retention [5,6]. As a result, the 36% of people with MS (pwMS) are unemployed, and 17% of workers with MS are forced into early retirement [6]. The premature job loss produces a financial burden for pwMS and their families [7,8], affecting individuals in the early stages of their professional careers [9]. In this context, work is also widely recognised as a pivotal social determinant of health, and an essential means of achieving self-determination and psychological well-being [10,11]. Ensuring that pwMS can access and retain their jobs is crucial not only to limit the economic burden but also to promote improved health management [10]. The diverse symptoms experienced by pwMS can result in a corresponding array of obstacles when attempting to access and maintain employment [12]. Architectural barriers in the workplace can impede the movement of workers with MS, both during the commute to and within the work environment and long consecutive working hours can lead to cognitive overload and extreme fatigue [12–14]. Administrative barriers, such as a lack of clarity regarding rights, have been shown to hinder access to social policies designed to support people with disabilities [15,16]. Furthermore, social barriers, such as judgemental work environments that demonstrate a lack of sensitivity to MS, have been demonstrated to engender severe stress and feelings of isolation [16]. Therefore, interventions aimed at facilitating the return to work, or job

placement require a multidisciplinary and person-centred approach to support pwMS facing the range of work-related difficulties they may experience [17–19]. According to Escorpizo et al., 2011, Vocational rehabilitation (VR) aims to optimise job participation in individuals with disability [20], and it has emerged as a promising method for delivering employment support services to individuals with different conditions [21]. In this framework, VR is conceptualised as an intervention capable of supporting people with disabilities at all stages of their working life, facilitating access in the workforce and job retention, as well as in the return to work [20]. VR interventions typically combine medical and rehabilitative components with strategies aimed at adapting the work environment to individual needs. In addition, these interventions may provide practical information and support to help individuals navigate administrative procedures and address barriers related to employment and social policies [12,22]. However, VR for pwMS is still an emerging field [22,23], and evidence regarding the efficacy of VR for pwMS is still limited, as highlighted in the Cochrane review by Khan et al., 2009, [3]. According to Munn et al., 2018 [24], we conducted a scoping review in order to identify the types of available evidence of VR for pwMS, examining how research is conducted, and to assess the need for a systematic review, highlighting possible barriers and facilitators that may arise in the delivery of VR interventions for pwMS. To this purpose, in addition to identifying the state of the art and characteristics of VR interventions, particular attention will be paid to feasibility studies and to stakeholders' opinions. Indeed, these two dimensions are now widely acknowledged as fundamental components in the definition of rehabilitation interventions, tailored to individuals' needs, and in the evaluation of VR interventions effectiveness [25,26].

## Aim and research questions

Specifically, we aim to summarise the main components of VR intervention, the professionals involved, the settings in which they are mainly conducted, the opinions of pwMS and the healthcare and social care professionals who participated in the VR interventions highlighting barriers associated with intervention delivery.

For this purpose, we formulated the following research questions:

1. What is the state of the art in vocational rehabilitation for pwMS?

   a. Where, when and how have vocational interventions been implemented and studied?

   b. What are the sociodemographic, clinical and occupational characteristics of the samples?

   c. What types of VR interventions are delivered, and which professionals are more likely to be involved in delivering vocational interventions?

2. What type of study designs are available in the literature?

3. What are the strengths and limitations highlighted so far in conducting vocational interventions for pwMS?

   a. What are the stakeholders' opinions (clients and health providers)?

   b. are the barriers and facilitators to implement a vocational intervention for pwMS?

## Scoping review methods

This scoping review was conducted following the Joanna Briggs Institute (JBI) recommendations for scoping reviews [27] and the Preferred Reporting Items for Systematic Reviews and Meta-Analyses extensions for Scoping Review (PRISMA-ScR) [28]. The protocol describing the scoping review process was previously registered on the Open Science Framework (OSF): 10.17605/OSF.IO/E3UF7

## Sources and search strategy

In order to identify articles, the following databases and registers were consulted: PubMed, SCOPUS, PsycINFO, CINAHL, Google Scholar, OT Seeker (University of Queensland), and the Physiotherapy Evidence Database (PEDro). An

ad hoc search strategy was developed for these sources, using two primary domains in accordance with the Population, Concept and Context (PCC) framework: one pertaining to the population of pwMS, and the other relating to the conceptual framework of the study, namely vocational rehabilitation. An iterative process was employed to optimise the search sensitivity, and the effectiveness of this process was tested multiple times to ensure its correct functioning. The selection of keywords and MeSH terms was informed by a comprehensive review of extant literature in the field and through deliberations with a multidisciplinary team with experience in VR and members of the Italian Multiple Sclerosis Society (AISM). Subsequently, according to Pollock et al., 2023 [29], the search strategy was adapted to align with the particular functionalities of each database and register that was consulted. A comprehensive search was conducted up to 6 October 2024. The records extracted were subsequently managed on Rayyan for title and abstract screening, as well as for the assessment of eligibility. Since 100% of the studies included were available on PubMed or Google Scholar, we re-ran the search on these two databases to identify any articles published while the review was in progress between 6 October 2024 and the end of the literature review phase on 22 October 2025.

The list of keywords and Mesh used in PubMed is reported in the S1 Table.

## Eligibility criteria

We included articles related to or derived from VR interventions in pwMS aged 18 years or older. In line with the definition proposed by Escorpizo et al. 2011, we included all types of VR interventions aimed at supporting pwMS in access to, retention in, and return to work. We included only peer-reviewed articles Grey literature was not explored. Only articles in English or Italian were considered as eligible. No filters related to the year of publication or study design were applied. No time limitation was set for the literature search.

## Exclusion criteria

We excluded articles that did not report separate data for pwMS or articles not related to VR interventions. We excluded articles generally focused on work and MS (e.g., barriers, risk factors) and articles related to VR interventions as main topic, but which do not follow the delivery of a VR intervention. Moreover, we excluded secondary sources (e.g., reviews) and articles published in language other than English or Italian.

## Study inclusion process

After consulting the databases and uploading the records obtained from the literature search, we eliminated duplicates. Once the final pool of articles had been identified, two independent reviewers (CG, FR) carried out the initial screening by consulting the title and abstract of each article. This phase was conducted according to the rule of "in case of doubt, keep it". Following the initial screening phases, a full-text inspection was conducted to ascertain their eligibility. The eligibility assessment was conducted by two independent reviewers (CG, FR). This phase was also conducted on Rayyan in order to facilitate the identification of conflicts between the two reviewers. Furthermore, a checklist developed specifically for the study in Excel was used to assess eligibility criteria and to track the reasons for exclusion for each article. Its appropriateness was tested by the research team conducting a pilot phase. In case of disagreement between the reviewers, a third reviewer inspected the full-text and resolved the conflicts (EG). Moreover, references of relevant articles and reviews were inspected to identify additional articles as well as a consultation with experts in the field. According to Munn and co-workers [24], no quality assessment was conducted.

## Data extraction

Data from the included studies were extracted by two reviewers (CG, FR) and reported in an Excel sheet. This file had been developed ad hoc for the purposes of this review. The appropriateness of the worksheet was evaluated during a pilot phase, with adjustments being made as necessary. According to Pollock et al. (2023) [29], a guideline related to the data

extraction sheet was created to standardise the process between the two reviewers. If available, the following information was extracted from articles, based on the PCC framework:

- Population: age, gender, education level, clinical status, marital status, EDSS and occupational status of individuals receiving intervention;

- Concept and context: the geographic locations of the interventions, the professionals who provided the intervention [e.g., occupational therapist, physiotherapist, neuropsychologist], the length of interventions, and the setting where it has been conducted. In addition, data on the study design and the methods employed to evaluate VR were extracted as well as a description of the VR programmes. If one article was related to a wider population including different diseases, we extracted only information and data related to pwMS. If articles described multiple interventions, we extracted separately information for each work-related intervention. Information on the implementation of VR interventions, including any barriers and facilitators, as well as stakeholder opinions was extracted, if available. In addition, other information on the feasibility of VR interventions for pwMS was extracted. Furthermore, the year of publication and the first author were documented.

### Data synthesis

The extracted data were summarised by one reviewer (CG) and reviewed by EG and FR. In accordance with Munn et al., 2018 [24] and Peters et al., 2021 [27], the extracted data were summarised qualitatively, without using qualitative data analysis methods such as thematic analysis. However, in order to provide an overview of the main characteristics of VR interventions, a classification of the main variables has been undertaken, according to the model described in Escorpizo et al., 2011 [20]. These variables include the type of intervention, the aim, the settings and the professionals involved. To classify VR programmes, the following macro areas of intervention were used as labels: rehabilitation, reasonable accommodation and education. According to Negrini et al., 2022 [30] and Escorpizo et al. 2011 [20], the label 'rehabilitation' was used for all interventions involving programmes of any kind of rehabilitation, such as occupational therapy, physiotherapy, neuropsychological therapy and psychological support, as well as courses for managing symptoms at work, peer-support and stress management interventions, support in communication and training of vocational skills. Those involving adjustments to workstations and/or working methods (e.g., smart working) were considered 'reasonable accommodation' (RA). Finally, VR interventions aimed at raising awareness and providing information to workers with MS about their rights and the difficulties they may experience in the workplace were categorised as "educational". Subsequently, interventions were classified as uni-dimensional if they encompassed only one of the three macro areas, or multi-dimensional if they involved at least two of the areas of intervention. In order to classify the aim of the intervention, we encompassed three distinct categories. First, interventions that support pwMS in seeking a job were classified as 'job access'. Second, interventions that offered assistance in returning to work following a diagnosis or hospitalisation were designated as 'job return'. Third, interventions aimed at maintaining the employment of already employed pwMS, enhancing work skills, and/or facilitating and optimising work participation were categorised as 'job retention'. Furthermore, we have distinguished the following settings: interventions conducted online, within vocational agencies, in inpatient or outpatient clinics.

Results were grouped and reported qualitatively in tables, according to the design employed in the studies that evaluated VR. This was done in order to follow the pyramid of scientific evidence, to highlight the level of evidence for VR that has been achieved to date. In instances where the same intervention was described in more than one article, we elected to include only one of the articles in order to calculate the frequency of key intervention characteristics (professionals involved, intervention components, settings). When more than one article was available for a given intervention, the most recent publication was included. Conversely, in instances where an article reported information pertaining to multiple interventions, the characteristics of each intervention were considered independently in the summary. Finally, opinions among

stakeholders were divided between feedback reported by pwMS and feedback from healthcare and social care professionals. Additionally, we distinguished the reported strengths from the perceived limitations. All descriptive analyses were performed using jamovi (version 2.6; The jamovi project, 2025).

## Results

The literature search identified 2,360 articles. After removing duplicates and screening, we included a total of 28 articles [16,31–57 35–42]. Details of the article inclusion process with reasons for exclusion are shown in Fig 1 according to PRISMA guidelines. When multiple exclusion criteria were met, all reasons for exclusion were recorded for each article at the eligibility stage.

### Characteristics of the included articles

The articles included were published between 1996 and 2025, covering approximately 30 years of research into interventions to support work activities for pwMS. The majority of the articles were conducted in Western countries, with a particular focus on United States (N = 10; 36%) [33,34,42,47–52,54] and on Europe, with studies conducted in Norway (N = 2; 7%) [32,44], Germany (N = 2; 7%) [38,45] Netherlands (N = 1; 4%) [31], Sweden [N = 1; 4%] [57] and Belgium (N = 1; 4%) [56]. The remain articles were conducted in the United Kingdom (N = 6; 21%) [16,35–37,43,53] and Australia (N = 4; 14%) [39–41,55]. A single study was conducted in Iran (N = 1; 4%) [46]. The designs adopted varied considerably between articles, but there has been an increase in interventional studies over the last 10 years of research. Five studies utilised a randomised (or quasi) controlled trial design, three of which assessed feasibility and acceptability of VR interventions

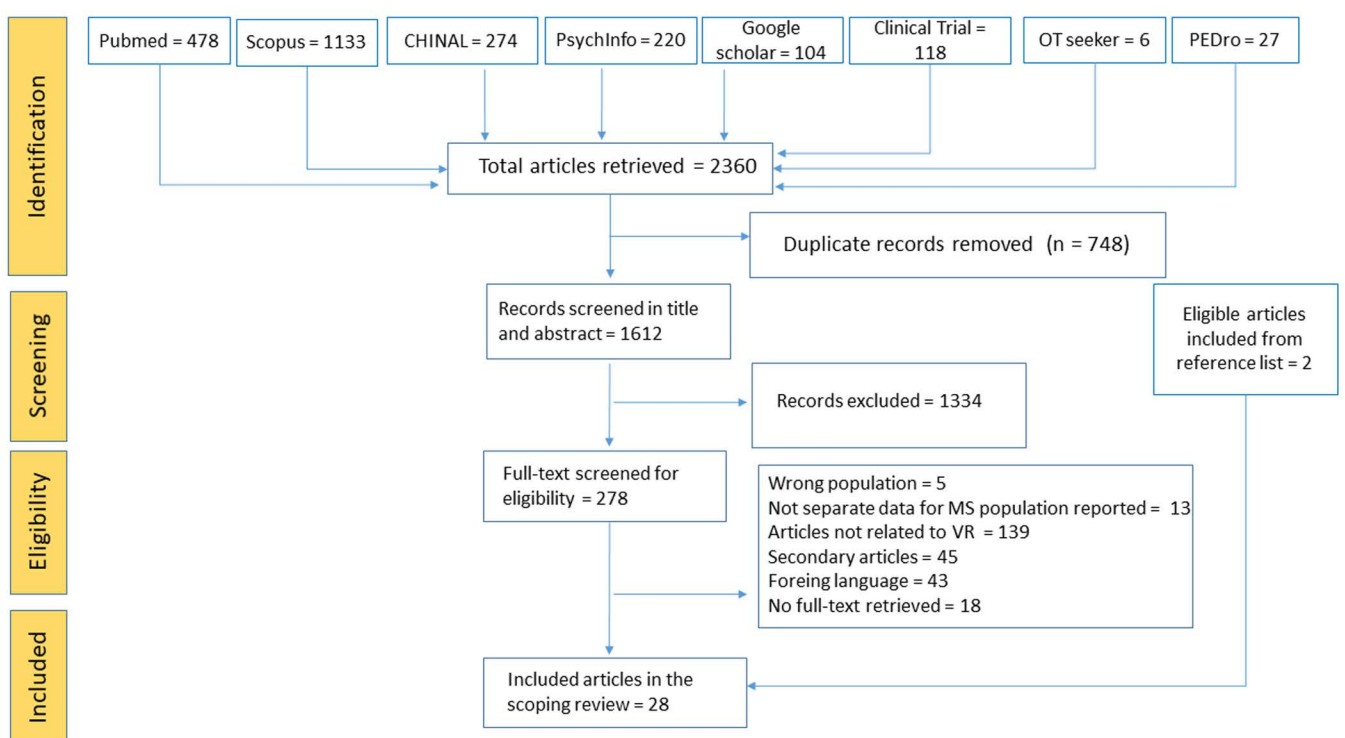

**Fig 1. Scoping review flowchart.** Flowchart according to PRISMA-ScR guidelines of the literature review process. We reported all the reasons for exclusion of each article evaluated in the eligibility phase.

(11%) [32,39,51] while two studies (7%) evaluated the efficacy of VR interventions [40,50]. The remaining articles utilised the following designs: non-RCT pilot study (N = 1; 4%) [41], observational (N = 6; 21%) [33,34,38,42,54,57], protocols (N = 5; 18%) [31,36,37,45,55], semi-experimental (N = 2; 7%) [46,48], mixed-methods (N = 4, 14%) [16,36,47,56], qualitative (N = 2; 7%) [43,44], case studies (N = 2; 7%) [52,53] and comparative (N = 1; 4%) [49].

Of the ten RCTs conducted or described within protocols, six (60%) adopted usual care as control group [31,32,36,50,51,55] while four (40%) reported waitlist control [39–41,45].

## Characteristics of the population

The majority of studies included pwMS without specifying eligibility criteria related to clinical status, MS phenotypes or age. However, a proportion of studies included participants with low or moderate levels of disability, while Aarts et al., 2024 [31] targeted pwMS with mild cognitive impairment and an EDSS < 6. One study included people with other neurological conditions [49]. Furthermore, two studies specifically targeted women with MS [50,51]. Four studies also included health professionals and mentors involved in the delivery of the intervention [16,35,37,39], and four studies included participants' employers [16,35–37]. All participants were over 18 years of age and under 64 years of age. The clinical, sociodemographic, and occupational characteristics of participants involved in VR programmes were reported heterogeneously across studies. Geographical origins or ethnicity of the samples were reported in eight articles [16,33–35,47,48,51,54]. In all of the eight studies, the samples predominantly comprised participants of European/Caucasian origin. Eleven articles reported the level of education of the sample [16,34,35,39–41,43,47,48,50,57]. In all of the eleven articles, the majority of participants held either a diploma or a university degree. Occupational status was reported in thirteen studies [16,32,33,35,38–41,47,50,53,56,57]. Most participants included in the studies were employed, with the exception of Chiu et al., 2015 [33] that included more unemployed participants. Marital status was reported in eight studies [16,35,39–41,47,50,51]. The majority of participants included were married or in a relationship. Disability levels were reported in five studies and measured with the Expanded Disability Status Scale (EDSS) [32,35,38,44,57], while MS phenotype was reported in eleven articles and disease duration in eight articles [16,32,35,38–41,43,53,57]. The samples consisted mainly of people with a relapsing-remitting (RR) MS phenotype, with low levels of disability.

Specific sociodemographic, clinical and occupational data related to the samples of the included articles are reported in Tables 1 and in S2. In cases where more than one cohort was present within an article, we reported the data separately.

## Characteristics of the vocational rehabilitation intervention included

The twenty-eight included articles documented a total of twenty-eight VR different interventions [31, 32, 33, 34, 38, 35, 36, 37, 16, 40, 39, 42, 43, 45, 46, 47, 48, 49_a, 49_b, 9_c, 50, 52; 53; 54, 55, 56, 57_a, 57_b]. Several articles reported more than one intervention, but the same intervention was reported in more than one article. Therefore, as detailed in the Methods section, only one study per intervention was included in this summary in order to avoid double counting the same intervention in the characteristics of the VR studies. In cases where multiple studies were available, the most recent study was selected. Wickstorm et al., 2017 [57] reported data from two interventions [57_a, 57_b], while Rumrill et al., 1996 [49] described four VR interventions: "The MS back to work", "the job raising program", "The return-to-work program" and "the career and possibilities project". However, one of the interventions reported in Rumrill et al., 1996, "the career and possibilities project", was also described in Rumrill et al., 1998. Harvedt et al., [2024] [44] reported the qualitative results of the coreDIST intervention, while Arntzen et al., [2023] [32] assessed its feasibility. Furthermore, Dorstyn et al. [2017] [41] and Dorstyn et al. [2018] [40] reported the results of the feasibility and effectiveness of the Work and MS package, respectively. VR interventions described by De Dios Perez and co-workers differ for the professional involved, the setting or the delivery methods, thus we consider them separately in the data synthesis [16,35–37].

Most of the VR programmes 61% [N = 17] were multi-dimensional in nature [34, 35, 36, 37, 16, 39, 42, 43, 45, 48, 49_a, 49_b; 49_c, 52, 53, 54, 55], with support being offered in more than one domains including rehabilitation, educational

**Table 1. Characteristics of the included articles. Data were reported separately for each cohort when a single article included more than one cohort.**

| Article | Year | Geographic area | Observation period | Design | Participants | N (Total)* | Workers group | F (n,%) | Age** (Total) | Main component of VR interventions |
|---|---|---|---|---|---|---|---|---|---|---|
| Aarts et al., [31] | 2024 | Netherlands | NA | RCT protocol | pwMS with mild cognitive impairment and EDSS < 6 | 270 | JR | NA | NA | Rehabilitation |
| Arntzen et al., [32] | 2023 | Norway | 2021 | Pilot RCT | pwMS with EDSS ≤ 3.5 | 29 | JR | 21, 72% | NR | Rehabilitation |
| Chiu et al., [33] | 2015 | USA | 2007-2011 | Case-control | pwMS | 8,715 | JB, JR, RTW | 5901, 68% | 16 - 64 | Reasonable accomodation |
| Chiu et al., [34] | 2013 | USA | 2009 | Cross-sectional | pwMS | 1920 | JB, JR, RTW | 1321, 69% | 42.6 (10.9) | Rehabilitation, reasonable accomodations and educational |
| De Dios Perez et al., [35] | 2025 | UK | 2022-2024 | Mixed-method feasibility | pwMS, employers and health professionals | 26 | JR | 13,65%*** | 48.6 (7.4)*** | Rehabilitation, reasonable accomodations and educational |
| De Dios Perez et al., [36] | 2025 | UK | 2025-2027 | RCT Protocol | pwMS and employers | 60 | JR | NA | NA | Rehabilitation, reasonable accomodations and educational |
| De Dios Perez et al., [37] | 2024 | UK | NA | Protocol | pwMS, employers and heatlh professionals | 30 | JR | NR | NA | Rehabilitation, reasonable accomodations and educational |
| De Dios Perez et al., [16] | 2023 | UK | 2020-2021 | Mixed-method case series | pwMS, employers and health professionals | 22 | JR | 12, 80%*** | 46.13 (9.58)*** | Rehabilitation, reasonable accomodations and educational |
| Dettmer et al., [38] | 2021 | Germany | 2019-2020 | Cross-sectional | pwMS | 64 | JS, JR, RTW | 43, 67% | 48.9 (8.7) | Rehabilitation |
| Dorstyn et al., [39] | 2022 | Australia | NR | Pilot RCT | pwMS and mentors | 29 | JS | 11, 79% | 45.4 (9.5) | Educational |
| Dorstyn et al., [40] | 2018 | Australia | NR | RCT | pwMS | 95 | JS | 81, 85% | 41.30 (9.8) | Educational |
| Dorstyn et al., [41] | 2017 | Australia | NR | Pilot study | pwMS | 29 | JS | 27, 93% | 44.4 (9.2) | Educational |
| Hartvedt et al., [44] | 2024 | Norway | 2021-2022 | Qualitative study | pwMS with mild to moderate disability | 26 | JR | 20, 23% | 48.5 (8.3) | Rehabilitation |
| Inge et al., [42] | 2016 | USA | 2011-2013 | Retrospective study | pwMS | 6,865 | JB, JR, RTW | NR | 25 - 64 | Rehabilitation, reasonable accomodations and educational |
| Jellie et al., [43] | 2014 | UK | NR | Qualitative study | pwMS | 19 | JB, JR, RTW | 15, 79% | 24 - 63 | Rehabilitation, reasonable accomodations and educational |
| Meyer-Moock et al., [45] | 2022 | Germany | 2022 | RCT Protocol | pwMS | NA | JR | NA | NA | Rehabilitation and educational |
| Nornematolahi et al., [46] | 2013 | Iran | NR | Semi-experimental | pwMS with EDSS grade 1–5 | 45 | JR | 45, 100% | 20 - 50 | Rehabilitation |
| Rumrill et al., [48] | 1998 | USA | NR | Semi-experimental | pwMS | 37 | JR | NR | NR | Reasonable accomodations and educational |
| Rumrill et al., [47] | 2013 | USA | 2001-2006 | Mixed-method | pwMS | 41 | JB, JR, RTW | NR | 46.9 (8.4) | Reasonable accomodation |

*(Continued)*

**Table 1.** (Continued)

| Article | Year | Geo-graphic area | Obser-vation period | Design | Participants | N (Total)* | Work-ers group | F (n,%) | Age** (Total) | Main component of VR interventions |
|---|---|---|---|---|---|---|---|---|---|---|
| Rumrill et al., [49] | 1996 | USA | NR | Comparative study | pwMS and other neurologic conditions | NA | RTW | NA | NR | Rehabilitation, rea-sonable accomoda-tions and educational |
| Stimmel et al., [50] | 2023 | USA | 2016-2019 | quasi-RCT | women with MS | 118 | JR | 118, 100% | NR | Rehabilitation |
| Stimmel et al., [51] | 2020 | USA | 2016-2017 | RCT pilot study | women with MS | 49 | JR | 49, 100% | 43 (10.4) | Rehabilitation |
| Strauser et al., [52] | 2019 | USA | NR | Case study | pwMS | 1 | RTW | 0% | 50 (0) | Rehabilitation, rea-sonable accomoda-tions and educational |
| Sweetland et al., [53] | 2014 | UK | 2009 | Case study | pwMS | 2 | RTW | 100% | 32 - 54 | Rehabilitation, rea-sonable accomoda-tions and educational |
| Tansey et al., [54] | 2015 | USA | NR | Quantitative descriptive | pwMS | 1920 | JB, JR, RTW | 1321, 69% | NR | Rehabilitation, rea-sonable accomoda-tions and educational |
| Van der Mei et al., [55] | 2024 | Australia | 2023-2024 | RCT protocol | pwMS | NA | JR | NA | NA | Rehabilitation and educational |
| Vonck et al., [56] | 2023 | Belgium | 2023 | Mixed-method | pwMS and employers | 223 | JB, JR, RTW | NR | NR | Educational |
| Wickstrom et al_North_a., [57] | 2016 | Sweden | 1995-2013 | Cohort study | pwMS living in north Sweden | 175 | JR | 116, 66% | 42 (11.5) | Rehabilitation |
| Wickstrom et al_South_b., [57] | 2016 | Sweden | 1995-2013 | Cohort study | pwMS living in south Sweden | 168 | JR | 120,71% | 44 (10.9) | Educational |

Legend: NA = Not applicable; NR = Not reported; RCT = Randomized Controlled Trial; JS = Job Seekers; JR = Workers in job retention; RTW = Workers returning to work *= number of subjects planned; ** = where available we extracted the mean and Standard Deviations age; in absence, we reported the range (r); ***= Data are related to pwMS.

interventions and reasonable accommodation. The remaining studies comprised uni-dimensional interventions [N = 11; 39%] [31, 32, 33, 38, 40, 46, 47, 50, 56, 57_a, 57_b] encompassing a single category of intervention among the three aforementioned categories. The most prevalent components within the VR intervention reported across the twenty-eight interventions were rehabilitation [N = 21; 75%] [31, 32, 34, 35_a, 36_b, 37, 15, 42, 43, 45, 46, 49_b, 49_c, 50, 52, 53, 54, 55, 57_a] and education (N = 19; 68%) [34, 35_a, 36_b, 37, 16, 40, 39, 42, 43, 45, 48, 49_a, 49_b, 49_c, 52, 54, 57_b], while reasonable accommodation was present in thirteen interventions (46%) [34, 33, 35_a, 36_b, 37, 16, 42, 43, 47, 48, 52, 53, 54].

The setting was documented in twenty-six of the interventions. The largest proportion of interventions were conducted in outpatient clinics (N = 11; 42%) [31, 32, 45, 48, 49_a, 49_b, 49_c, 50, 53, 57_a, 57_b] followed by seven studies that were conducted online (27%) [39, 40, 36_b, 37, 16, 55, 56], five within vocational agencies (19%) [33,34,42,47,54] and two (8%) in inpatient clinics [38,43]. Finally, De Dios Perez et al., 2025_a [35] conducted the intervention in a hybrid for-mat (online and face to face) (N = 1; 4%) at the clinical rehabilitation centre according to the participant's preference.

The professionals involved were reported in nineteen interventions. The professional figure with the highest level of involvement was the psychologist, who was present in eight of the interventions described (42%) [16,37,39,40,43,45,50,55], followed by the occupational therapist (N = 4; 21%) [35,37,43,53] and the physiotherapist (N = 3; 16%) [32,43,46]. Other professionals involved in the VR programmes were: vocational agency professionals (N = 5; 26%) [33,34,42,47,54] specialist doctors (N = 2; 11%) [32,45] and MS nurses (N = 1; 5%) [32]. In addition, other non-health professionals such as MS charities employees or social insurance suppliers and mentors (peer-support) (N = 4; 21%) were reported [31,36,39,45].

The interventions implemented were predominantly focused on job retention (N = 13; 46%) [31, 32, 35, 36, 37, 16, 45, 46, 50, 52, 55, 57_a, 57_b], while nine (32%) interventions were also directed towards job access or return to work following a prolonged absence [34, 54, 38, 42, 43, 47, 49_b, 54, 56]. The proportion of interventions that were directed exclusively towards job seekers, whether employed or unemployed, was found to be 14% (N = 4) [39, 40, 48, 49_a]. Finally, Sweetland et al., 2014 [53] intervention was directed towards both the reintegration into work and the job maintenance [N = 1; 4%].

Finally, 7 out of 28 (25%) interventions adopted a goal-oriented approach [16,31,35–37,43,48], and fourteen (46%) interventions reported an individualised programme [16,31–37,42,43,45,48,53,54].

## Rehabilitation programmes

Across the twenty-one rehabilitation programmes, we identified seven types of intervention: symptom management, functional rehabilitation, stress management, work–life balance, support in communicating the diagnosis, assertive communication training, peer support, and vocational skills training. Below, we report the frequency of each of these seven interventions within the twenty-one rehabilitation programmes identified. Each rehabilitation programme included one or more of these seven types of intervention. Of the twenty-one rehabilitation components of VR intervention, twelve interventions were aimed at supporting pwMS in managing their symptoms in the workplace (N = 12; 57%) [31, 34, 35, 36, 37, 16, 42, 45, 49_c, 50, 54, 55]. These interventions provided strategies to overcome symptom-related difficulties in the workplace and enhance awareness of the impact of MS on work ability. The rehabilitation of functions, encompassing physiotherapy, occupational therapy, neuropsychological rehabilitation and psychological support, was documented in eight out of twenty-one of the included VR interventions (38%). Two of these interventions provided physical activity [32] and aerobic training in water [46]. The other interventions provided rehabilitation based on individual needs, without starting from a predefined programme or protocol [34, 42, 43, 45, 54, 57_a]. The remaining interventions were targeted towards the facilitation of communication in the workplace (N = 9; 43%) [34, 42, 43, 45, 54, 57_a], the disclosure of the diagnosis (N = 10; 48%) [34, 35, 36, 37,16, 42, 49_b, 53, 54, 55] and in the vocational skills training (N = 9; 43%) [32, 34, 16, 35, 36, 37, 42, 49_b, 54]. Finally, a number of interventions were designed to assist pwMS in the alleviation of stress and mood difficulties, or the enhancement of coping mechanisms (N = 6; 29%) [35, 36, 37, 16, 49_b, 52], in achieving a balance between professional engagements and their daily lives (N = 5; 24%) [31, 34, 42, 54, 55], or were grounded in the concept of peer support (N = 3; 14%) [31, 39, 49_c].

## Reasonable accomodations (RA) programmes

We identified two different types of RA interventions within the thirteen VR intervention that comprised RA: interventions aimed at providing advice on workplace adaptations and interventions aimed at providing accommodation. These two different types of RA may coexist within the same VR interventions, or they may be present individually. Of the thirteen VR programmes which included a RA interventions, all studies provided an evaluation of the workplace or a consultations for adjustments and specific aids (100%) [16,33–37,42,43,47,48,52–54], while five (39%) directly provided the accommodation needed by the participant [33,34,42,47,48,54]. RA were mainly evaluated from VR agency databases, especially the studies in which RA has been fully provided [33,34,42,47,54]. VR agency provides the following kinds of RA schedule

modification: equipment/assistive technology (e.g., enlarged computer screen, scooter, voice activated software), climate control/Air conditioning, reassignment, reduction of hours, physical accessibility [47].

### Educational programmes

Within the educational programmes, we have identified the following four areas of intervention: legal support, information on the labour market, referral to other sources of information, and career development support. Each educational intervention may contain more than one of these components. Of the nineteen educational programmes, the majority were aimed at providing information on legal rights (N = 11; 58%) [35; 36; 37; 16; 40; 39; 43; 45; 52; 55; 57_b]. Ten interventions (53%) were aimed at supporting pwMS in defining their career [35, 36, 37, 16, 40, 39, 48, 49_a, 49_c, 55] and nine interventions (47%) referred to other resources to support people with disability experiencing work-related difficulties [35_a, 36_b, 37, 16, 40, 39, 49_a, 52, 56]. Finally, six educational programmes (31%) provided information on the job market to support pwMS in access to new job position [40, 39, 48, 49_a, 49_b, 57_b].

The main findings and intervention features are reported in Table 2. Fig 2 shows the main interventions for each of the VR dimensions for pwMS.

### Stakeholder opinion: Perceived barriers and facilitators to the intervention

The opinions of stakeholders were reported in seven articles [16,35,41,43,44,48,51]. We summarised the main aspects in Table 4. All studies focused on the perspective of pwMS, while the opinion of professionals was only investigated by De Dios Perez et al., 2023 [16] and De Dios Perez et al., 2025 [35], who also included the opinion of employers within the VR intervention. All seven studies reported positive aspects and strengths of VR programmes, as well as some limitations.

PwMS reported that the adoption of an open-minded and sensitive approach by healthcare providers was paramount. This approach allows engendering a sense of understanding and acknowledgement [16,35,43]. The most beneficial initiatives in pwMS's opinion also included awareness courses on the rights of workers with disability, information regarding the management of symptoms, empowerment, and guidance on entering the job market [16,43,51]. Similar results were also reported by Dorstyn et al., 2017 [41], where the provision of bespoke multimedia content, customised to individual case narratives and accompanied by guidelines on the disclosure of illness within the workplace, the provision of generic information on job-seeking and the employment market was found to be very helpful. Furthermore, Jellie et al., 2014 [43] also highlighted the importance of the support in the management of coping mechanisms for anxiety and loss of confidence. Stimmel et al., 2020 [51], where personalised guidance concerning cognitive functioning and in-person feedback following neuropsychological assessments were pivotal to the VR intervention.

De Dios Perez et al.,2025 [35] highlights a strong acceptance of remote vocational support, valued for its flexibility, personalisation, and clinical relevance. Benefits were greatest when the intervention was timely and delivered within supportive organisational environments.

In Hartvedt et al., 2024 [44] physical activity (PA) was perceived to improve work capacity, performance, satisfaction and retention. PA was identified as a pivotal factor in the domains of identity, self-esteem and quality of life. It is evident that the psychological benefits of PA, including enhanced tolerance and strength, have contributed to the ability of individuals to manage mental challenges and stress. However, these strengths have been reported as a limitation by other participants, who perceive PA as an activity that consumes energy and difficult to integrate with work.

Finally, in Rumrill et al., 1998 [48] pwMS reported that the professional suggestions received, the identification of accommodation needs, and the support in the accommodation assessment with employers were central in their VR programmes.

In terms of limitations, pwMS reported a preference for in-person meetings as opposed to online meetings, in instances where the subject matter pertains to emotionally sensitive aspects of their work. As Dorstyn et al. [2022] [39] asserted, the online format has also been identified as a potential barrier to participation. However, in contrast to these findings,

**Table 2. Description of interventions reported in the included studies stratified by study design. If one article described more than one intervention, we reported each intervention separately.**

| Article | Year | Setting | Duration of intervention | Control group or comparison | Name of intervention | Description | Professionals involved | Aim (of intervention) | Primary outcome | Main findings |
|---|---|---|---|---|---|---|---|---|---|---|
| **Case studies** | | | | | | | | | | |
| Strauser et al., [52] | 2019 | NR | NR | NA | The Illinois work and well-being model | The Illinois Work and Well-Being Model is oriented toward career development for pwMS, based on contextual domains, considering personal and environmental factors. Practitioners may provide direct services, vocational counselling, self-advocacy training, job accommodation strategies, referral to other specialists to manage symptoms and stress | NR | Job retention | NA | Effective interaction between multiple domains were seen as a key aspect in VR |
| Sweetland et al., [53] | 2014 | Outpatients centers | 9 h | Usual care | NR | This intervention consists of assessment of needs, support with diagnosis disclosure at work, environmental evaluation of workplace, meetings to discuss options for individuals and employers | Occupational therapist | Job return and retention | NR | Participants were effectively reintegrated into the workforce after the intervention early support and education on workplace disability management at work alleviated concerns and worries about pwMS |
| **Qualitative studies** | | | | | | | | | | |
| Hartvedt et al., [44] | 2024 | MS outpatient clinic and Municipality | 10 weeks | Usual care | The coreDIST | The coreDIST participation includes interlinked phases involving digital communications with an MS nurse regarding work difficulties, needs and potential facilitators; a physiotherapy assessment, and physiotherapist-driven outdoor sessions conducting Group-CoreDIST exercises, balance and gait training; consultations with the employer regarding work, physical activity and potential adaptation | MS nurse, physiotherapist and generalists | Job retention | Feasibility of physical activities within the workplace and the perceived effect on work abilities | Physical activities in the workplace are feasible, and employers evaluate them as effective for self-perception and work capabilities |

*(Continued)*

**Table 2.** (Continued)

| Article | Year | Setting | Duration of inter-vention | Control group or com-parison | Name of interven-tion | Description | Profession-als involved | Aim (of interven-tion) | Primary outcome | Main findings |
|---|---|---|---|---|---|---|---|---|---|---|
| Jellie et al., [43] | 2014 | University hospital | NR | NA | NR | This intervention consists of a goal-oriented intervention with an individualised plan, which could comprise rehabil-itation and education on legal obligations. Depending on their needs, participants may be referred to neuropsychology, physiotherapy, and the Access to Work Scheme. Furthermore, they receive information related to their rights. When neces-sary, a visit to the workplace is undertaken to identify possible workplace adjustments | Neuropsy-chologist, Physioter-apist and Occu-pational therapist | Job access, return and retention | Participants prospec-tive of VR interventions | Participants reported the VR interventions as effective, in partic-ular for understanding the impact of MS on work-related abilities and for support in man-aging them |
| **Descriptive and comparative study** | | | | | | | | | | |
| Inge et al., [42] | 2016 | VR agency | NR | NA | NR | VR services included assess-ment of needs, VR counsel-ing and guidance, diagnosis and treatment, rehabilitation technology, job placement assistance, transportation, job search assistance, informa-tion and referral, on-the-job supports, maintenance training, college or university training, occupational/vocational training, job readiness training, technical assistance, disability-related augmentative training, on-the-job training, supported employment, personal attendant services | VR agency professionals | Job access, return and retention | Employment outcomes achieved by pwMS | Within the study period, approximately 2,006 cases (representing 29% of the total case closures of individuals with MS) were closed with an employment out-come. The mean weekly hours of employment for PwMS were approxi-mately 26.5 hours, and the mean weekly earn-ings were approximately $440. The mean case service expenditure per person with MS who achieved employment outcomes ranged from $8,372 to $9,066. |

*(Continued)*

| Article | Year | Setting | Duration of intervention | Control group or comparison | Name of intervention | Description | Professionals involved | Aim (of intervention) | Primary outcome | Main findings |
|---|---|---|---|---|---|---|---|---|---|---|
| Tansey et al., [54] | 2015 | VR agency | NR | NA | NR | VR services included assessment to evaluate the individual employment plan, diagnosis and treatment of impairments, VR counselling and guidance, college or university training, occupational/vocational training, on-the-job training, basic academic, remedial or literacy training, job readiness training, augmentative skills training, miscellaneous training, job search assistance, job placement assistance, on-the-job support (coaching), training and support to use transportation services, maintenance (monetary and basic necessities), rehabilitation technology | VR agency professionals | job access, return and retention | Service utilization and difference in service utilization between job seekers and employed pwMS | Job seekers are more likely to receive services focused on job orientation and training, while employed pwMS received career stabilisation interventions such as assistive technology or accommodation services. These services, provided by VR federal agencies, reduce the disparity between the average worker in the USA and the outcome achieved by VR agencies |
| Rumrill et al_a., [49] | 1996 | Patient foundation and research university center | NR | NR | The MS back to work | The MS Back to Work – Operation Job Match focuses on supporting job placement of pwMS, enhancing their proficiency in job seeking, and providing assistance from the employment community to develop a variety of career options. It is part of the Job Raising Programme | NR | Job access | NR | The success of this initiative lies in its adaptability. It has been conducted and replicated in numerous locations of the Society Chapter nationwide |
| Rumrill et al_b., [49] | 1996 | Patient foundation and research university center | 10 weeks | NR | The Job Raising Program | The Job Raising Program provides direct services and information pointing to: assertiveness, interviewing skills, resume writing, job market, diagnosis disclosure, self-assessment strategy and mutual support, and stress and coping management | NR | job access, return and retention | NR | The networking between disability groups, rehabilitation centres and employers was the most important part of the intervention. |

*(Continued)*

**Table 2.** (Continued)

| Article | Year | Setting | Duration of intervention | Control group or comparison | Name of intervention | Description | Professionals involved | Aim (of intervention) | Primary outcome | Main findings |
|---------|------|---------|--------------------------|------------------------------|----------------------|-------------|------------------------|------------------------|-----------------|---------------|
| Rumrill et al_c., [49] | 1996 | Patient foundation and research university center | NR | NR | The return-to-work program | The return-to-work program is a small-group intervention which intends to support pwMS in overcoming disability benefit restrictions and employer stereotypes regarding pwMS, and motivate participants in return to work after acute phase and support them in accepting diagnosis. They also were assigned to mentors expert in their job field | NR | Job return | NR | The programme provides assistance to individuals who elect to re-enter the workforce. The provision of timely informational and educational support is of the utmost importance. |
| Rumrill et al_d., [49] | 1996 | MS clinics and | NR | NR | The career possibilities project | The career possibilities project involves consultations via telephone with a rehabilitation specialist regarding the participants' vocational goals and job experiences. Following the conclusion of consultations, participants are presented to an employer within their field of interest with a view to facilitating their access to employment. Furthermore, a half-day programme is administered with the objective of assisting PwMS in identifying the accommodations they require. | NR | Job access | NR | The Career Possibilities Project seminar has been shown to engender increased confidence in the job search process, as well as in the frequency of job-seeking activities. Furthermore, it has been demonstrated to engender a greater sense of optimism with regard to the future of the career of unemployed pwMS. |
| **Mix-methods** | | | | | | | | | | |
| Vonck et al., [56] | 2023 | Online | NR | NA | Online MS toolkit | The Online MS Toolkit comprises two elements: firstly, a screening tool designed to detect and monitor work difficulties over time; and secondly, a dashboard to facilitate the swift and straightforward identification of information related to work difficulties for people living with MS. The document is available for consultation by both professionals and pwMS. | NR | job access, return and retention | NA | MS toolkit: screening tool and dashboard |

*(Continued)*

**Table 2.** (Continued)

| Article | Year | Setting | Duration of intervention | Control group or comparison | Name of intervention | Description | Professionals involved | Aim (of intervention) | Primary outcome | Main findings |
|---|---|---|---|---|---|---|---|---|---|---|
| Rumrill et al., [47] | 2013 | VR agency | NR | NA | NR | Reasonable accommodations provided include: schedule modification; equipment/assistive technology (e.g., enlarged computer screen, scooter, voice activated software); Climate control/Air conditioning Reassignment/Reduction of hours; ADA/Disclosure information; Physical accessibility | VR agency professionals | job access, return and retention | Current employment status and accommodation experience | 60% of pwMS who had received a rasonable accomodation were still working from 10 to 15 years after adjustments. The low cost-accomodations were the most frequently implemented by employers |
| De Dios Perez et al., [16] | 2023 | Online | 12 weeks | NA | NR | Understanding MS, education about legal rights, support with disclosure, fatigue management, signposting to local and national resources, advice about RA, employer engagement, cognition in MS, MS and emotions, long-term career planning, referrals | Psychologist | Job retention | Feasibility: recruiting, time to recruit the sample, practical attrition and dropouts reasons, appropriateness of methods and procedures | The intervention was feasible to deliver. There was a significant positive impact on goal attainment immediately following VR |
| De Dios Perez et al., [35] | 2025 | Neurology servise in NHS hospital | 12 weeks | NA | NR | Understanding MS, education about legal rights, support with disclosure, fatigue management, signposting with local and national resources, advice about RA, employer engagement, cognition in MS, MS and emotions, long-term career planning, referrals | Occupationa therapist | Job retention | Feasibility was assessed by recruitment rates, compliance, and practicality of delivery. Acceptability was assessed with post-intervention interviews | Recruitment and training an OT was challenging. Factors affecting intervention adherence included annual leave and family responsibilities. VR was associated with improved vocational goal attainment post-intervention and at follow-up |

*(Continued)*

**Table 2.** (Continued)

| Article | Year | Setting | Duration of intervention | Control group or comparison | Name of intervention | Description | Professionals involved | Aim (of intervention) | Primary outcome | Main findings |
|---|---|---|---|---|---|---|---|---|---|---|
| **Cross-sectional** | | | | | | | | | | |
| Chiu et al., [34] | 2013 | VR agency | NA | NA | NR | VR services included assessment of needs, VR counselling and guidance, diagnosis and treatment, rehabilitation technology, job placement assistance, transportation, job search assistance, information and referral, on-the-job supports, maintenance training, college or university training, occupational/vocational training, job readiness training, technical assistance, disability-related augmentative training, on-the-job training, supported employment, personal attendant services, reader services, and interpreter (no language) services. | VR agency professionals | job access, return and retention | Employment status: competitive employment | VR services are positively associated with pwMS who were successfully employed after intervention. |
| Dettmer et al., [38] | 2021 | Neurological inpatients clinics | 3 to 4 weeks | NA | NR | VR as an add-on to the standardised programme of physiotherapy, occupational therapy, speech pathology and psychological therapy | NR | job access, return and retention | subjective fatigue | Cognitive fatigability, measured with Tonic Alertness, can predict employment status better than self-perception of fatigability, measured with the FSMC, 3 months after rehabilitation |
| **Case-control study** | | | | | | | | | | |
| Chiu et al., [33] | 2015 | VR agency | NR | pwMS who does not received rehabilitation | NR | Rehabilitation technology services (assistive technology and job accommodations) offered by VR agencies in the USA. | VR agency professionals | job access, return and retention | Employment status: competitive employment | Rehabilitation technology is an effective intervention to promote job retention outcomes among workers with MS |
| **Cohort study** | | | | | | | | | | |
| Wickstrom et al North_a., [57] | 2016 | outpatients centers | 3 weeks | pwMS treated in the south of Sweden | NR | A 3-week long comprehensive rehabilitation course, with a follow-up week after 6 months. This course equips the patients with work-promoting tools. | NR | Job retention | Work ability | Proportions of pwMS who remain in the workforce were higher in the north compared with southern populations |

*(Continued)*

| Article | Year | Setting | Duration of intervention | Control group or comparison | Name of intervention | Description | Professionals involved | Aim (of intervention) | Primary outcome | Main findings |
|---|---|---|---|---|---|---|---|---|---|---|
| Wick-strom et al_South_b., [57] | 2016 | outpatients centers | 6 weeks | pwMS treated in the north of Sweden | NR | A brief information course consisting of a weekly 3 hour session during a period of 6 weeks, in the early stages of the disease. | NR | Job retention | Work ability | Proportions of pwMS who remain in the workforce were higher in the north compared with southern populations |
| **Semi-experimental** | | | | | | | | | | |
| Nornem-atolahi et al., [46] | 2013 | NR | 8 weeks | NR | NR | This intervention consists of an aquatic aerobic training with intensity of 50–60 of maximum heart rate, 3 sessions a week. | Physiotherapist | Job retention | work-related quality of life | Water aerobic exercis-esignificantly improved quality of work life and in the experimental group and improved the mean score of quality of work life in pwMS on the average of 9.28 per cent. Doing selected water aerobic exercise improves the qualityof work life and in pwMS |
| Rumrill et al., [48] | 1998 | MS society chapter | NR | Active control | The Career Possibilities Project | The Career Possibilities Project involves the administration of telephone consultations with a rehabilitation specialist, with the aim of ascertaining the participants' vocational goals and past employment experi-ence. Following the conclusion of the consultations, participants are presented to an employer in their field of interest with a view to facilitating their access to employment. Furthermore, a half-day programme is administered with the objective of assisting PwMS in identify-ing the accommodations they require | NR | Job access | Accomodation self-efficacy, employability maturity and employment status | 11 out of 37 pwMS were successfully re-entered within the workforce. |
| **Protocols** | | | | | | | | | | |
| Aarts et al., [31] | 2024 | outpatients clinics | 4 months | Enhaced usaul care | strength-ening the mind | The Strengthening the Mind is a work-focused intervention com-bining the capability approach and the participatory approach in one-on-one coaching by trained work coaches who have MS themselves | Life-style coach and trained work mentors | Job retention | Quality of life | NA |

*(Continued)*

| Article | Year | Setting | Duration of intervention | Control group or comparison | Name of intervention | Description | Professionals involved | Aim (of intervention) | Primary outcome | Main findings |
|---------|------|---------|--------------------------|-----------------------------|----------------------|-------------|------------------------|-----------------------|-----------------|---------------|
| De Dios Perez,, [36] | 2025 | Online | 6 months | Usual care | NR | This goal-oriented intervention is based on the individual's needs. It comprises separate programmes for participants and employers. Topics covered for people with MS include: Understanding MS; Advice on reasonable adjustments; Support for requesting reasonable adjustments; Fatigue management; Managing cognition at work; Information about legal rights; Diagnosis disclosure; Long-term career planning; Managing mood difficulties; Signposting to local and national resources.For employers: Signposting to relevant organisations; Educational resources; Information about MS and invisible symptoms; Support with identifying reasonable adjustments; Legal responsibilities under the Equality Act | MS charities employees | Job retention | NR | NA |
| De Dios Perez,, [37] | 2024 | Online | 12 weeks | NA | NR | This intervention involves a three-month job retention intervention including an initial interview, vocational goal setting, and up to 10 h of individually tailored support dependent on the needs and/or goals of the person with multiple sclerosis. Education about multiple sclerosis, legal rights, support with disclosure, symptoms management, advice about reasonable adjustments, employer engagement, sign posting to local and national resources, managing emotions, long-term careerplanning, and referrals to other services | Occupational therapist and/or psychologist | Job retention | NR | NA |

*(Continued)*

| Article | Year | Setting | Duration of intervention | Control group or comparison | Name of intervention | Description | Professionals involved | Aim (of intervention) | Primary outcome | Main findings |
|---|---|---|---|---|---|---|---|---|---|---|
| Meyer-Moock et al., [45] | 2022 | outpatients clinics | 24 months | Waitlist | MSnetWork-study | This intervention is based on participant's needs. Depending on subjective work-related needs, the neurologist can activate different programs: disease education, rehabilitation of functions, psychosocial and legal counselling | Neurologists, occupational health and rehabilitation physicians, psychologists, and social insurance suppliers | Job retention | Number of sick leave days | NA |
| Van der Mei et al., [55] | 2024 | Online | 10 weeks | Usual care | The MS WorkSmart | The MS WorkSmart intervention package includes access to MS WorkSmart, access to the My SymptoMS app, coaching support via Zoom/telephone, and access to a closed Facebook group. MS WorkSmart consists of nine modules: 1) General introduction to MS and work; 2) Symptom management; 3) Effective communication; 4) Disclosure diagnosis at work; 5) Effectively communicating factual information; 6] Transformative shift in participants' perspective on work challenges: 7) Mindful living; 8) Modification of the workplace; 9) Plan for future | Coach-psychologyst trained in deliver Cognitive Behavioural Therapy | Job retention | Feasibility and acceptability of intervention measured with recruitment and eligibility data, retention and missing on data collection, acceptability of intervention areas and adherence | NA |

**Feasibility and Pilot (RCT study)**

| Article | Year | Setting | Duration of intervention | Control group or comparison | Name of intervention | Description | Professionals involved | Aim (of intervention) | Primary outcome | Main findings |
|---|---|---|---|---|---|---|---|---|---|---|
| Arntzen et al., [32] | 2023 | outpatients clinics | 8 weeks | Usual care | The coreDIST participation | The coreDIST participation foresees interlinked phases involving digital communications with a MS nurse regarding work difficulties, needs and potential facilitators; a physiotherapy physical assessment, and physiotherapist-driven outdoor sessions conducting Group-CoreDIST exercises pointing to balance and walking training; consultations with the employer regarding work, physical activity and potential adaptation | MS nurse, physiotherapist and generalists | Job retention | Feasibility of the intervention in terms of process, resources, management, and scientific outcomes | The primary feasibility metric outcomes demonstrated the need for minor adjustments in regard to the content of the intervention, and increasing the number of staff. |

*(Continued)*

| Article | Year | Setting | Duration of intervention | Control group or comparison | Name of intervention | Description | Professionals involved | Aim (of intervention) | Primary outcome | Main findings |
|---|---|---|---|---|---|---|---|---|---|---|
| Dorstyn et al., [41] | 2017 | Online | 6 weeks | Waitlist | The Work and MS package | The Work and MS Package addresses the social, psychological, legal, and medical aspects connected to employment. It consists of seven modules: one introductory module and six instructional modules. These materials includes links to vocational and psychological support resources, government regulations, job-search websites, and mobile health applications. Each module also offered optional homework activities, and the possibility to receive email-based guidance from both a moderator and a psychologist | Moderators and Psychologist | Job access | Vocational self-efficacy and identity, life orientation and mood | The findings provide preliminary support for the utility of a job-information resource, 'Work and MS', to augment existing employment services. Significant effects are reported for vocational self-efficacy, identity, and optimism |
| Dorstyn et al., [39] | 2022 | Online | 8 weeks | Waitlist | The MS JobSeek | The MS JobSeek consists of 7 online information modules and discussions with peer mentors. Modules cover the following themes: Information on MS and Job; Finding the right job; Resume writing and application letter; Successful job interview; Maintaining work; Career development and planning | Mentors and psychologist | Job access | Feasibility: forum engagement and satisfaction and job search behaviour, efficacy and quality of life | Findings support the feasibility of the intervention. However, attrition ates were reported (43%) and no group or time effect is reported. pwMS valued the utility of the program |
| Stimmel et al., [51] | 2020 | Tertiary care MS center | 12 months | Usual care | NR | This intervention comprises a neuropsychological assessment for employed women. After neuropsychological assessment, in person feedback and case management is provided | Psychologist | Job retention | Feasibility (enrolment and attrition rates) and acceptability (benefits form interventions) of VR and evaluations of symptoms change | 97% rated the intervention as beneficial and the intervention was feasible. However, there was a high drop-out rate (6 out of 22) |

*(Continued)*

**Randomized (or quasi) Controlled Trial pointing to efficacy**

| Article | Year | Setting | Duration of intervention | Control group or comparison | Name of intervention | Description | Professionals involved | Aim (of intervention) | Primary outcome | Main findings |
|---|---|---|---|---|---|---|---|---|---|---|
| Dorstyn et al., [40] | 2018 | Online | 4 weeks | Waitlist | The Work and MS package | The Work and MS Package covers social, psychological, legal, and medical needs related to work. The package contains seven PDF documents: an introductory module and six learning modules. These included links to vocational and psychological resources, government legislation, jobseeking webpages, and mobile healthcare applications. Optional homework tasks were incorporated into the modules, with opportunity to access email-based support from a moderator and a psychologist | Moderators and Psychologist | Job access | Vocational interest and self-efficacy in job-seeking activities | pwMS who accessed the work and MS intervention reported an improved confidence in their career goals |
| Stimmel et al., [50] | 2023 | Tertiary care MS center | 12 months | Usual care | NR | This intervention comprises a neuropsychological assessment for employed women. After neuropsychological assessment, in person feedback and case management is provided | Psychologist | Job retention | Maintenance of employment status, treatment recommendations adherence and symptoms improvement | Employment status did not differ between the experimental and control groups, but adherence to treatment was significantly higher in the experimental group |

**Legend**: NA = Not applicable; NR = Not reported; pwMS = people with multiple sclerosis; VR = Vocational Rehabilitation; PA = Physical activity.

**Fig 2. Main components of vocational rehabilitation for pwMS.** The figure summarizes the principal domains of VR interventions identified in the scoping review. Interventions were grouped into three main areas: Rehabilitation, reasonable accommodations, and educational.

Stimmel et al. [2020] [51] reported that pwMS cited travelling to the clinic as a barrier to programme participation, and in most cases, a reason for withdrawal. Furthermore, in Dorstyn et al., 2017 [41], pwMS reported a lack of provision of information regarding legal issues, strategies for maintaining a healthy work-life balance, but also the need for an effective method to interact with employers. Finally, the involvement of employers could prove invaluable; however, this must be approached with the utmost caution by the participants [16].

Healthcare professionals believed that there is a need to provide VR to PwMS, while employers reported an increased understanding of the needs of their employees with MS [16]. Although integration into NHS pathways was considered valuable, several practical and financial constraints were identified [35].

### Feasibility of interventions

Six interventional studies investigated the feasibility of the interventions [16,32,35,39,41,51]. Feasibility was assessed differently across the five studies, but all reported data on enrolment and subject retention in the study. In general, the included articles highlighted a good recruitment capacity and a variable adherence to the proposed activities. Only Stimmel et al., [2020] [51] contacted a large number of eligible participants, but included only a small percentage of them. Participation was occasionally constrained by logistical, motivational or health-related impediments, but drop-outs were minimal and predominantly ascribed to personal or medical justifications [32,35,39,41]. Although certain authors documented response rates that fell below the anticipated figures, the general feasibility of the programmes was deemed sufficient, with the potential for future implementation in larger settings [16,32,35,39,41,51].

We have summarised the main results for the reported variables of the workflow of the intervention in Table 4.

### Discussion

To our knowledge, this is the first comprehensive scoping review aiming to identify the entire body of literature related to VR interventions to support pwMS to access, return to or to maintain job position. Previous systematic and scoping reviews aimed to assess the efficacy of the intervention, including only a few articles and without providing a landscape of the VR programmes for pwMS worldwide [3,22,23]. Thus, this is the first scoping review to encompass both qualitative and feasibility data related to the implementation of VR for pwMS. Our results suggest that a substantial corpus of

**Table 3. Synthesis of stakeholders' perspectives on their participation in VR interventions. The main strengths and limitations reported by pwMS, professionals, and employers are presented.**

| Articles | years | Strength - pwMS | Limits - pwMS | Strength – Professionals and employers | Limits – Professionals |
|---|---|---|---|---|---|
| De Dios Perez et al., [16] | 2023 | • The content and structure of the VR were evaluated as acceptable by the participants.• Open-minded and sensitive approach of healthcare providers, and expertise in legal aspects• The most beneficial measures identified were awareness courses on the rights of workers with disability, information regarding the symptoms, management, empowerment, and guidance on entering the job market | • Preference for in-person meetings in cases of emotionally sensitive aspects of work.• The involvement of employers could be precious, but it has to be carefully evaluated by the participants | • The healthcare professionals believed that there was a need to provide VR for people with MS.• After the intervention, employers understood the needs of their employees with MS better | • Unwareness of pwMS regarding the impact of MS in their work life |
| De Dios Perez et al., [35] | 2025 | • Access to a type of support not previously available• Valued the flexibility and reduced burden of remote delivery• Perceived the approach as more personalised than standard NHS• Empathic, knowledgeable, and trustworthy approach of HCP• Substantial benefit when support is aligned with personal timing• Substantial benefits in supportive organisations, including an enhanced sense of value and more effective implementation of adjustment• Strong interest in having such support embedded within NHS care | • Barriers to discussing work difficulties within routine care• Providing support too early or too late risked overwhelm or disengagement• In unsupportive workplaces, adaptations were difficult to implement• Concerns about discrimination reduced engagement and intervention impact• Risk of fragmented provision or reliance on voluntary sector organisation | • Improved understanding of MS and employees' needs, supported by practical information• Perception that employees gained confidence and agency when negotiating work adaptations• HCPs Recognize vocational support as clinically relevant.• Remote delivery is seen as facilitating adherence and continuity and the importance of the intervention timing• Recognition that organisational culture shapes vocational and clinical outcomes | • Effectiveness diminished when workplaces were unwilling or unable to engage collaboratively• Uncertainty about funding organisational placement within the NHS• Potential difficulties for non-specialist OTs• Ethical constraints on contacting workplaces directly• Practical challenges in balancing research or vocational tasks with routine NHS responsibilities• Structural and financial constraints |
| Dorstyn et al., [41] | 2017 | • Tailored multimedia addressing personal case stories and guidelines on the disclosure of illness in the workplace.• Generic information on job-seeking and the employment market was considered very helpful.• Participants would recommend the intervention to their peers | • Online tools and information result in barriers for some older pwMS.• Add more information regarding legal issues, how to maintain a healthy work-life balance, and how to effectively interact with employers not prepared to accommodate for disability | NA | NA |
| Hartvedt et al., [44] | 2024 | • Online tools and information result in barriers for some older pwMS.• Add more information regarding legal issues, how to maintain a healthy work-life balance, and how to effectively interact with employers not prepared to accommodate for disability | • For some participants PA was considered energy-consuming and limiting.• Difficulties in balancing daily life PA with work | NA | NA |

*(Continued)*

**Table 3.** (Continued)

| Articles | years | Strength - pwMS | Limits - pwMS | Strength – Professionals and employers | Limits – Professionals |
|---|---|---|---|---|---|
| Jellie et al., [43] | 2014 | • Understanding my symptoms and their management in the workplace• Managing anxieties• Better coping with the employer• Managing loss of confidence• Receiving professional support and empathic and proactive approach | NR | NA | NA |
| Rumrill et al., [48] | 1998 | • Interview suggestions, the identification of accommodation needs, orientation, accommoda-tion evaluation with employers and placement planning | NR | NA | NA |
| Stimmel et al., [51] | 2020 | • Perceived benefits of neuropsy-chological feedback• Person-alised recommendations• Value of follow-up support | • Manage the excessive workload and related stress by participating in the interven-tion.• Reach the location of the intervention due to excessive distance | NA | NA |

**Legend**: NA = Not applicable; NR = Not reported; pwMS = people with multiple sclerosis; VR = Vocational Rehabilitation; PA = Physical activity.

research has been undertaken over time, highlighting VR as a highly sensitive issue for pwMS. However, at this juncture, the timing is not conducive to undertaking a systematic review, since high-quality RCTs are still lacking. However, the presence of several interventional protocols with randomization and control group has been reported recently and herald the arrival of highly scientific results [31,36,45,55].

The included studies were mainly conducted in Western countries. Nevertheless, the increasing prevalence of MS in non-Western countries emphasises the importance of conducting these studies worldwide [58,59]. Furthermore, the scop-ing review evidenced considerable heterogeneity in the reporting of sociodemographic and occupational characteristics of study samples, as well as in the description of work-related outcomes, across studies on VR in multiple sclerosis. In line with previous recommendations [60], we call for established guidelines to assist authors in the reporting of sociodemo-graphic and occupational data, to enhance the representation of this population within research and to allow the stratifica-tion of clinical studies results based on these characteristics as well. This is particularly crucial to facilitate meta-analyses that encompass factors impacting the efficacy of VR intervention and to inform a systematic review [24].

The present review, in accordance with Escorpizo and colleagues'[20] definition, identified studies aimed at supporting pwMS throughout all stages of working life, conducted not solely by vocational agencies but also in outpatient or inpatient settings. Furthermore, Escorpizo et al., [2011] [20] emphasise the importance of addressing individual factors hindering participation in work, together with environmental factors. In accordance with this definition, the articles included pertained to multi-dimensional interventions that addressed work-related difficulties from multiple perspectives. We identified three main categories of VR intervention: rehabilitation, reasonable accommodation and educational interventions. Rehabil-itation was the most prevalent component within VR interventions. In the VR context, rehabilitation aimed at assisting individuals in managing symptoms in the workplace and in identifying effective strategies to overcome work-related chal-lenges. Furthermore, programmes that focus on enhancing communication skills, particularly in the context of disclosing diagnoses, are essential components of VR programmes [43,61,62]. Indeed, the process of diagnosis has been shown to engender a range of psychological effects in pwMS, including the questioning of professional capabilities and the devel-opment of persistent negative thought patterns and maladaptive coping strategies [11,52,62,63]. Consequently, recent

**Table 4. Feasibility of VR interventions. The main results from feasibility and pilot interventional studies are reported.**

| Article | Year | Assessed for eligibility (n) | Excluded (n) | Reason for non eligibility | Included (Interventional group) | Included (Control Group) | Drop out – intervention group | Drop out – control group | Reasons for drop-outs or missing appointment |
|---|---|---|---|---|---|---|---|---|---|
| Arntzen et al., [32] | 2024 | 35 | 6 | Not meeting inclusion criteria; declined to participate (withdraw of consent, lack of time, other personal reasons) | 15 | 14 | 2 | | knee injury and illness |
| De Dios Perez et al., [16] | 2023 | 26 | 4 | Not meeting inclusion criteria; declined to participate; other reasons | 22 | NA | All participants who started the intervention completed it, and there were no dropouts | NA | NA |
| De Dios Perez et al., [35] | 2025 | 36 | 16 | 1 = ineligible because he/she did not have an official diagnosis of MS. The other 15 were not recruited for reasons unrelated to eligibility. Did not respond to email = 8; Limited availability = 2; Interested but recruitment closed = 5 | 20 | 0 | 1 | 0 | One participant dropped out after the initial assessment because he/she started a new job and no longer had time. |
| Dorstyn et al., [39] | 2022 | 43 | 14 | Not meeting inclusion criteria; did not complete baseline survey; contact details not provided | 14 | 15 | 0 | 0 | Two subjects were lost to follow-up and reported occurring health issues; and one third have found a new job position and did not. They were still included in the final analysis |
| Dorstyn et al., [41] | 2017 | NR | NR | NR | 29 | NA | 11 | NR | Reason for drop-outs were not routinely assessed. However, reason for withdrawal were mainly related to time contraits, occurring health issues, and due to change job condition |
| Stimmel et al., [51] | 2020 | 628 | 579 | 316 = unemployment; 37 = retirement; 7 = no MS diagnosis; Not approached = 123; Declined to participate = 30; did not meet cut-off on screening measures = 22 | 16 | 22 | 6 | 8 | NR |

Legend: VR = Vocationa Rehabilitation; pwMS = people with Multiple Sclerosis; NA = Not applicable; NR = Not reported; RCT = Randomized Controlled Trial; OT = Occupational therapist.

studies have identified the importance of timely intervention following diagnosis as a means of mitigating the impact on vocational identity, self-esteem and self-efficacy [53,62,64]. Timing is crucial: premature intervention can be overwhelming, whereas intervention initiated at a later stage may become ineffective, due to the exacerbation of work-related difficulties [35]. In this regard, the most suitable figures to promptly identify work-related needs are primary care professionals, such as general practitioners or specialist nurses. Addressing work-related difficulties within primary care settings enables pwMS to engage with these issues when they feel ready, in a trusted environment, thereby reducing the risk of delayed intervention. In this light, the provision of bespoke training for professionals in the workplace is indispensable, a necessity that is not only expressed by professionals involved in VR programmes but is also a key demand [65,66].

Furthermore, the presence of communication difficulties is indicative not only of impaired psychological functioning and reduced well-being, but also of concomitant common symptoms such as cognitive impairment and dysarthria [67]. This underscores the multidisciplinary nature of VR interventions which is characterised by the diversity of intervention domains within VR, and by multidisciplinary intricacies inherent within these domains. Collaborative efforts to effectively address the multifaceted challenges posed by work-related difficulties are needed [18,64], and this is reflected in the significant involvement of psychologists and occupational therapists in the VR programmes reported in the included studies.

A distinctive feature of VR interventions is the general absence of eligibility restrictions based on disease phenotype, disease duration or level of disability across the included studies. In this review, only two VR interventions selected participants with an average level of disability: the CoreDist participation [32,44] and the Strengthening the Mind [31]. While these findings should be interpreted with caution, they indicate that VR interventions have been applied in heterogeneous pwMS populations. In line with previous literature [13,68], work-related difficulties may occur across different forms and stages of multiple sclerosis, suggesting that individuals may benefit from VR interventions regardless of their clinical profile or level of disability. The inclusion of diverse clinical profiles may also reflect an underlying reliance on bio-psycho-social frameworks, which conceptualise work-related functioning as the result of interacting physical, cognitive, and contextual factors rather than disease severity alone [64,69].

Despite goal-oriented interventions being widely adopted in MS rehabilitation [70], our review identified only a few studies with a collaborative approach aimed at tackling specific work-related objectives [16,31,35,36,43,48]. De Dios Pérez et al., [2023] [16] highlighted the difficulty that pwMS experience in recognising the impact of their illness on their ability to work, and the subsequent challenge of accepting this impact. Conversely, a person-centred approach involving the development of personalised rehabilitation plans is more commonly adopted in VR for pwMS [16,33,34,37,42,43,45,48,53,54,64]. In this context, we believe that a person-centred approach and goal-oriented interventions, combined with individualised rehabilitation plans, are the most effective tools for addressing the variety of work-related difficulties and clinical characteristics experienced by people with MS [13,35–37,64].

Educational interventions are also frequently used, either as a standalone program or in combination with rehabilitation and reasonable accommodation. Indeed, the complexity of administrative procedures governing access rights for persons with disabilities has been demonstrated to act as a significant impediment to employment [13,15] for both those seeking to maintain their current position and those aspiring to transition to a new role [39–41]. In this particular context, the review identified several interventions that resulted in the creation of information packages with references to existing employment support resources [39,40,55,56].

Reasonable accommodation (RA) was the least frequent component within VR programmes. Nevertheless, the necessity to intervene at an environmental level is recognised as an essential element in ensuring that individuals with MS remain in work [12,13,47]. Furthermore, VR interventions that directly provided accommodation were primarily conducted using data from the databases of US vocational agencies [33,42,47,54]. Consequently, there is a paucity of evidence of effectiveness conducted outside the US and within healthcare settings. Research conducted outside vocational agencies has provided advice on accommodations, but there has been no subsequent follow-up on the provision of accommodations by employers. Further studies are needed to assess the effectiveness of RA, as well as their feasibility. Thus, this

review reveals that the social context of work and relationships with employers can hinder the direct implementation of RA [35,37]. Without a supportive environment, accepting aids and adjustments can be challenging as it necessitates making one's disability visible, exacerbating social isolation. In these contexts, pwMS may refuse the intervention, in order to avoid the deterioration of their relationships with colleagues and employers [17,35,37].

VR interventions included in our review were assessed as feasible and acceptable [16,32,39,41,51]. However, the reasons for ineligibility or withdrawal from the study were not systematically reported across studies and several barriers have been highlighted within this scoping review such as: proximity to the centre providing the service, lack of time due to excessive workload [51]; lack of time for personal reasons [44] and, in interventions aimed at changing job position or to enter the workforce, pwMS may change work position, or find the new job prior to receiving or completing the intervention or due to health issues [39,41]. These results highlight the need to design interventions that can be easily adapted to personal needs in terms of delivery methods and timing, adapting service schedules as far as possible outside peak working hours in order to facilitate participation, especially for those who are already employed and face work-related difficulties [61]. Furthermore, we did not identify studies evaluating the cost-effectiveness of VR interventions, which is a key aspect for the implementation of the service within the NHS.

No cross-cutting preference for online interventions emerged: in De Dios Perez et al., 2023 [16], participants reported that addressing sensitive issues in person would have been more satisfactory for them, and in Dorstyn et al., 2017 [41], online intervention proved difficult for older people. PwMS recognised the importance of specific interventions aimed at tackling work-related difficulties, and evaluated an empathetic, non-judgemental and welcoming approach as essential to support them in accessing employment [16,43]. However, more studies are needed to frame stakeholder opinions, especially with regard to professionals and employers, who were only included in the study by De Dios Perez et al. (2023) [16].

## Conclusions

We have mapped the existing literature on VR for pwMS who need to enter, return to or remain in employment. Further high-quality scientific studies are needed in order to identify the most effective interventions and ensure a pathway that meets the real needs of workers. The implementation of VR interventions at both the environmental and individual levels introduce a heightened level of complexity, necessitating a more sophisticated scientific evaluation. However, these interventions possess the capacity to address authentic needs and are evaluated as acceptable and feasible.

### Strengths and limitations

This is the first comprehensive scoping review of VR interventions for pwMS. This study examined the implementation process and considered the opinions of the relevant stakeholders, which is an innovation in the field of scoping review. A plethora of databases was consulted to include all the relevant publications; nevertheless, we refrained from utilising more extensive keywords associated with VR, such as "work," in order to circumvent literature that encompasses work-related challenges without offering any insights pertaining to VR. Furthermore, we did not request the full texts from the authors of the articles if they were not available elsewhere. This may have excluded some articles of possible interest. Finally, the data on the study populations were characterised by a high degree of heterogeneity, thus we were not able to provide percentages related to the sample characteristics.

### Supporting information

**S1 Table. Search Strategy.** Search strings used for each electronic database, including MeSH terms and free-text keywords.
(DOCX)

**S2 Table. Sociodemographic, clinical and occupational data.** Detailed characteristics of participants across the included studies, including, geographical origins, marital and occupational status, MS type, EDSS and disease duration.
(DOCX)

**S3 Table. Data extraction.** Full data extraction form including articles information, population and intervention characteristics, and stakeholder opinions.
(XLSX)

**S4 Prisma-ScR Checklist. PRISMA-ScR Checklist.** Preferred Reporting Items for Systematic Reviews and Meta-Analyses extension for Scoping Reviews (PRISMA-ScR) checklist.
(DOCX)

## Author contributions

**Conceptualization:** Carlotta Gualco, Erica Grange, Marco Salivetto, Michela Ponzio.

**Data curation:** Carlotta Gualco, Federica Rotondi.

**Formal analysis:** Carlotta Gualco, Michela Ponzio.

**Funding acquisition:** Michela Ponzio.

**Investigation:** Carlotta Gualco, Federica Rotondi.

**Methodology:** Carlotta Gualco, Erica Grange, Michela Ponzio.

**Project administration:** Carlotta Gualco, Michela Ponzio.

**Resources:** Carlotta Gualco.

**Software:** Carlotta Gualco.

**Supervision:** Carlotta Gualco, Michela Ponzio.

**Validation:** Carlotta Gualco.

**Visualization:** Carlotta Gualco.

**Writing – original draft:** Carlotta Gualco.

**Writing – review & editing:** Carlotta Gualco, Erica Grange, Federica Rotondi, Marco Salivetto, Elena Pignattelli, Tommaso Manacorda, Maria Grazia Grasso, Giorgia Presicce, Matilde Inglese, Lorenza Nasone, Paolo Durando, Guglielmo Dini, Benedetta Persechino, Giampaolo Brichetto, Michela Ponzio.

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
