## [Decision Letter · Decision Letter 0]

3 May 2026

PONE-D-26-06438Vocational Rehabilitation for people with Multiple Sclerosis: A Systematic Scoping Review of International EvidencePLOS One

Dear Dr. Gualco,

Thank you for submitting your manuscript to PLOS ONE. After careful consideration, we feel that it has merit but does not fully meet PLOS ONE’s publication criteria as it currently stands. Therefore, we invite you to submit a revised version of the manuscript that addresses the points raised during the review process.

We look forward to receiving your revised manuscript.

Kind regards,

Karlo Toljan

Academic Editor

PLOS One

Journal Requirements:

We note that one or more of the authors is affiliated with the funding organization, indicating the funder may have had some role in the design, data collection, analysis or preparation of your manuscript for publication; in other words, the funder played an indirect role through the participation of the co-authors. If the funding organization did not play a role in the study design, data collection and analysis, decision to publish, or preparation of the manuscript and only provided financial support in the form of authors' salaries and/or research materials, please do the following:

1. Review your statements relating to the author contributions, and ensure you have specifically and accurately indicated the role(s) that these authors had in your study. These amendments should be made in the online form.

2. Confirm in your cover letter that you agree with the following statement, and we will change the online submission form on your behalf:

“The funder provided support in the form of salaries for authors [insert relevant initials], but did not have any additional role in the study design, data collection and analysis, decision to publish, or preparation of the manuscript. The specific roles of these authors are articulated in the ‘author contributions’ section.

Additional Editor Comments:

Dear authors,

I appreciate your patience. Unfortunately, it was difficult to secure reviewers sooner.

Reviewers' comments:

Reviewer's Responses to Questions

**Comments to the Author**

1. Is the manuscript technically sound, and do the data support the conclusions?

Reviewer #1: Partly

2. Has the statistical analysis been performed appropriately and rigorously? 

Reviewer #1: N/A

3. Have the authors made all data underlying the findings in their manuscript fully available?

Reviewer #1: Yes

4. Is the manuscript presented in an intelligible fashion and written in standard English?

Reviewer #1: Yes

5. Review Comments to the Author

Reviewer #1: I enjoyed reading this paper and I like the scope of this review including a summary of stakeholders’ perspective. I do have some concerns, which are listed below:

• There should be one clear definition of vocational rehabilitation that is used throughout the article. The authors use different definitions in the abstract (‘Vocational rehabilitation aims to optimize job access and the return to work through multidisciplinary and person-centered approaches.’, ‘Studies were eligible if they were related to VR interventions for pwMS, focused on job access, return, or retention and if they were primary articles’) and introduction (‘Vocational rehabilitation (VR) aims to ease work difficulties and to optimise work participation in individuals with chronic illness or disability’)

• The systematic review from Kahn and colleagues is very relevant for this topic and should be mentioned in the article: Effectiveness of vocational rehabilitation intervention on the return to work and employment of persons with multiple sclerosis - PubMed

• It should be clear in table 1 who the target group for the intervention is since the authors include VR interventions for people who are working, who are looking for work etc. and these groups are very different

• The descriptive statistics in the text describe which articles included different characteristics of the population (e.g. ethnicity) but the article does not specify this further; I’d like to see this either in the text or one of the tables (e.g. what ethnicities were included and in which articles)

• The authors conclude that there is a pivotal role of VR interventions in facilitating access to and retention in work for PwMS, however effects were not taken into account in this review so that statement feels out of place

6. PLOS authors have the option to publish the peer review history of their article (what does this mean?). If published, this will include your full peer review and any attached files.

Reviewer #1: **Yes:**Shalina R.D. Saddal

---

## [Author Response · Author response to Decision Letter 1]

7 May 2026

Author’s Response

Reviewer specific comments and responses

1. There should be one clear definition of vocational rehabilitation that is used throughout the article. The authors use different definitions in the abstract (‘Vocational rehabilitation aims to optimize job access and the return to work through multidisciplinary and person-centered approaches.’, ‘Studies were eligible if they were related to VR interventions for pwMS, focused on job access, return, or retention and if they were primary articles’) and introduction (‘Vocational rehabilitation (VR) aims to ease work difficulties and to optimise work participation in individuals with chronic illness or disability’)

#1: We would like to thank the reviewer once again for giving us the opportunity to clarify the theoretical background on which this scoping review is based. Throughout the scoping review, our intention was to consistently use the definition provided by Escorpizo et al (2011). To avoid any confusion regarding differing definitions, we have revised the abstract and main text to ensure consistent use of a single definition of vocational rehabilitation throughout the manuscript and clarified all relevant passages accordingly.

In the abstract, VR interventions are now described as follow:

According to Escorpizo et al., 2011 framework, Vocational rehabilitation (VR) aims to optimise job participation, providing support in the job access, retention and in the return to work for people with disability.

In the introduction:

According to Escorpizo et al., 2011, Vocational rehabilitation (VR) aims to optimise job participation in individuals with disability [20], and it has emerged as a promising method for delivering employment support services to individuals with different conditions [21]. In this framework, VR is conceptualised as an intervention capable of supporting people with disabilities at all stages of their working life, facilitating access in the workforce and job retention , as well as in the return to work [20]

In the eligibility criteria:

In line with the definition proposed by Escorpizo et al. 2011, we included all types of VR interventions aimed at supporting pwMS in access to, retention in, and return to work.

2. The systematic review from Kahn and colleagues is very relevant for this topic and should be mentioned in the article: Effectiveness of vocational rehabilitation intervention on the return to work and employment of persons with multiple sclerosis - PubMed

#2: We would like to thank the reviewer for their suggestion. The Cochrane systematic review by Khan et al. (2009) has been discussed as a relevant background source in both the Introduction (74 and 101-102) and in the Discussion (453-454). It is included as Reference 3 in the reference list. To improve clarity and facilitate identification, we have checked the citation and ensured that the full reference is accurately reported in the manuscript.

In addition, we have included a more explicit reference to the review in the introduction (lines 101–102), as follows:

However, VR for pwMS is still an emerging field [22, 23], and evidence regarding the efficacy of VR for pwMS is still limited, as highlighted in the Cochrane review by Khan et al., 2009, [3].

3. It should be clear in table 1 who the target group for the intervention is since the authors include VR interventions for people who are working, who are looking for work etc. and these groups are very different

#3: We would like to thank the reviewer for this comment, which we believe will improve the qualitative summary of the results. We have therefore added a new column to Table 1, entitled ‘Worker characteristics’, in which we have included three possible categories: ‘Job seekers’, ‘Workers needing job retention support’ and ‘Workers returning to work’. All the changes can be seen in the version “Revised Manuscript with Track Changes”.

4. The descriptive statistics in the text describe which articles included different characteristics of the population (e.g. ethnicity) but the article does not specify this further; I’d like to see this either in the text or one of the tables (e.g. what ethnicities were included and in which articles)

#4: Thank you for this helpful comment. Detailed study-level information regarding participant sociodemographic and population characteristics (including, where reported, ethnicity/geographical origin, education, marital status, and occupational variables) is provided in the Supplementary Materials (Table S2). Given the heterogeneity in the reporting of these variables across studies, we chose to summarise the descriptive statistics narratively in the main text while presenting the full study-level details in the supplementary materials to preserve readability and avoid overburdening the main manuscript. In the narrative synthesis, we reported only the most frequently represented category within each variable (e.g., predominantly Caucasian/European participants where ethnicity was reported) to provide an overall summary of sample characteristics. To improve clarity, we have revised the manuscript to more explicitly indicate that detailed participant characteristics for each included study are available in supporting information (S2 Table. Detailed characteristics of participants across the included studies, including, geographical origins, marital and occupational status, MS type, EDSS and disease duration). All the changes can be seen in the version “Revised Manuscript with Track Changes”.

5. The authors conclude that there is a pivotal role of VR interventions in facilitating access to and retention in work for PwMS, however effects were not taken into account in this review so that statement feels out of place

#5: We would like to thank the reviewer for pointing out this inconsistency. We agree that this sentence goes beyond the findings and scope of this scoping review. We have therefore amended the conclusions, changing the sentence as follows (All the changes can be seen in the version “Revised Manuscript with Track Changes”):

We have mapped the existing literature on VR for pwMS who need to enter, return to or remain in employment. (…)

---

## [Editor Report · Decision Letter 1]

11 May 2026

Vocational rehabilitation for people with multiple sclerosis: a systematic scoping review of international evidence

PONE-D-26-06438R1

Dear Dr. Gualco,

We’re pleased to inform you that your manuscript has been judged scientifically suitable for publication and will be formally accepted for publication once it meets all outstanding technical requirements.

Kind regards,

Karlo Toljan

Academic Editor

PLOS One
---

## [Editor Report · Acceptance letter]

PONE-D-26-06438R1

PLOS One

Dear Dr. Gualco,

I'm pleased to inform you that your manuscript has been deemed suitable for publication in PLOS One. Congratulations! Your manuscript is now being handed over to our production team.

Kind regards,

on behalf of

Dr. Karlo Toljan

Academic Editor

PLOS One